# Dinuclear Iron Complexes of Iminopyridine-Based Ligands as Selective Cytotoxins for Tumor Cells and Inhibitors of Cancer Cell Migration

**DOI:** 10.3390/pharmaceutics14122801

**Published:** 2022-12-14

**Authors:** Jessica Castro, Marlon Bravo, Meritxell Albertí, Anaís Marsal, María José Alonso-De Gennaro, Oriol Martínez-Ferraté, Carmen Claver, Piet W. N. M. van Leeuwen, Isabel Romero, Antoni Benito, Maria Vilanova

**Affiliations:** 1Laboratori d’Enginyeria de Proteïnes, Departament de Biologia, Facultat de Ciències, Universitat de Girona, Campus de Montilivi, C/Maria Aurèlia Capmany, 40, 17003 Girona, Spain; 2Institut d’Investigació Biomèdica de Girona Dr. Josep Trueta, Parc Hospitalari Martí i Julià—Edifici M2 C/Dr. Castany s/n, 17190 Salt Girona, Spain; 3Departament de Quimica Física e Inorgànica, Universitat Rovira i Virgili, Campus Sescelades, 43007 Tarragona, Spain; 4Institut Català d’Investigació Química (ICIQ), Avinguda dels Països Catalans, 16, 43007 Tarragona, Spain; 5Departament de Química and Serveis Tècnics de Recerca, Universitat de Girona, Campus de Montilivi, C/Maria Aurèlia Capmany, 69, 17003 Girona, Spain

**Keywords:** dinuclear iron (II) compounds, iminopyridine ligands, selectivity for tumor cells, cytotoxic mechanism, apoptosis, inhibition of cell migration

## Abstract

A family of dinuclear iron (II) compounds with iminopyridine-based ligands displays selective cytotoxic activity against cancer cell lines. All compounds have IC_50_ values 2–6 fold lower than that of cisplatin, and 30–90 fold lower than that of carboplatin for the tumor cell lines assayed. Comparing the IC_50_ values between tumor and non-tumor cell lines, the selectivity indexes range from 3.2 to 34, compound **10, [Fe_2_(4)_2_(CH_3_CN)_4_](BF_4_)_4_**, showing the highest selectivity. Those compounds carrying substituents on the iminopyridine ring show the same cytotoxicity as those without substituents. However, the electronic effects of the substituents on position 6 may be important for the cytotoxicity of the complexes, and consequently for their selectivity. All compounds act over DNA, promoting cuts on both strands in the presence of reactive oxygen species. Since compound **10** presented the highest selectivity, its cytotoxic effect was further characterized. It induces apoptosis, affects cell cycle phase distribution in a cell-dependent manner, and its cytotoxic effect is linked to reactive oxygen species generation. In addition, it decreases tumor cell migration, showing potential antimetastatic effects. These properties make compound **10** a good lead antitumor agent among all compounds studied here.

## 1. Introduction

The use of metals in medical applications dates back to ancient civilizations [1]. However, it was not until the last century that the important therapeutic activity of metal complexes was discovered, with the outstanding finding of the anticancer activity of cisplatin [2]. From then on, the scientific community has embarked on a widespread search for related compounds with the same activity but fewer and less serious side effects. Second and third generation of cisplatin analogues have been developed, but, unfortunately, they still suffer from persistent, non-desired effects and intrinsic or acquired tumor drug-resistance. Thus, new metal complexes have been explored as anticancer agents, among them iron complex compounds [3]. These iron complexes are especially attractive, since iron is present in living systems and therefore shows low toxicity to normal cells. In the human body, iron is involved in important biological processes, such as electron and oxygen transport, usually in complex forms incorporated into target proteins such as hemoproteins (hemoglobin or myoglobin) or non-heme proteins (flavin-iron enzymes, transferrin, and ferritin) [4]. Interestingly, iron compounds with antitumor properties act through several anticancer mechanisms that differ from those used by platinum compounds [3]. The iron compounds with anticancer properties that have been most studied are organometallic complexes of ferrocene, and mononuclear coordination complexes of iron(II) and iron(III) [5,6]. In the latter group, iron (II) and (III) polypyridyl complexes, salen- and salophen iron derivatives, thiosemicarbazone, and iron-nitrosyl complexes, have been widely studied [6]. In general, these mononuclear compounds containing different ligands are selectively cytotoxic for several human cancer cell lines. Some of them display activities several fold higher than that of cisplatin [3].

Among the different metal compounds with antitumor activity described in the literature, multi-nuclear compounds present additive effects of their individual metal centers, which endow them with great potential as anticancer drugs. In addition, multi-nuclear compounds show novel DNA binding modes that are distinct from cisplatin. Among them is found long range DNA intrastrand or interstrand crosslinking ability, bisintercalation, major or minor groove binding, and electrostatic binding [7]. Indeed, multi-nuclear platinum compounds show very good cytotoxicity against different human tumor cell lines [8]. For instance, triplatin (BBR3464) has entered clinical phase II trials [9]. Several bimetallic mixed (organometallic/coordination) complexes that contain a ferrocene and different transition metals, such as Au, Cu, Pd, Pt, Rh, and Ru, have been synthesized to study their anticancer properties. They show improved cytotoxicity compared with the corresponding ferrocenyl motifs [10]. 1D-polymeric iron (III) salen-like complexes containing N-donor heterocyclic ligands, such as imidazole, 1,2,4-triazole, benzotriazole, 5-methyltetrazole, 5-aminotetrazole, and 5-phenyltetrazole, show cytotoxic activity against a panel of tumor cell lines. Unfortunately, the compounds were not stable, and, actually, the calculated IC_50_ values were the result of a mixture of compounds and reactants [11]. Unlike polymeric and mixed bimetallic iron compounds, few dinuclear iron compounds have been tested as antitumor agents. Some of them, such as [Cl(HPClNOL)Fe(μ-O)Fe(HPClNOL)Cl]Cl_2_·2H_2_O, and [(SO_4_)(HPClNOL)Fe(μ-O)Fe(HPClNOL)(SO_4_)] · 6H_2_O, where (HPClNOL = 1-(bis-pyridin-2-ylmethyl-amino)-3-chloropropan-2-ol), have enhanced nuclease activity, but, when tested as antitumor agents, are not highly cytotoxic [12]. On the other hand, the antitumor properties of nitrosyl iron complex with sulfur-containing ligand 2-aminophenol-2-yl, a structural analog of DNA thiopyridine base [13], and those of dinuclear neutral sulfur-nitrosyl iron complex with 2-mercaptobenzthiazole as a ligand [Fe_2_(C_7_H_4_NS_2_)_2_(NO)_4_] [14], have also been studied. Both compounds are cytotoxic for different tumor cell lines inducing apoptosis linked to the release of nitrogen oxide (NO). There is a consensus on the need to develop new therapeutic agents against cancer, since there is not a general guideline towards the synthesis of new active metal compounds, and the treatment of cancer remains a challenge.

Using iminopyridine ligands based on a methanodibenzodioxocine (DBDOC) backbone (Figure 1), a series of dinuclear iron compounds were synthesized, **[Fe_2_1_2_(CH_3_CN)_4_](BF_4_)_4_ 7, [Fe_2_2_2_(CH_3_CN)_4_](BF_4_)_4_ 8**, **[Fe_2_3_2_(CH_3_CN)_4_](BF_4_)_4_ 9, [Fe_2_4_2_(CH_3_CN)_4_](BF_4_)_4_ 10, [Fe_2_5_2_(CH_3_CN)_4_](BF_4_)_4_ 11,** and **[Fe_2_6_2_(CH_3_CN)_4_](BF_4_)_4_ 12.** Some of their crystal structures were determined by X-ray crystallography (Figure 2) [15,16]. These compounds have been studied as catalysts in the cycloaddition of CO_2_ to epoxides to obtain cyclic carbonates [15], but have not been tested as antitumor agents. Our hypothesis is that these dinuclear iron (II) complexes containing iminopyridine-based ligands (Figure 1) with different substituents on the pyridyl rings could have different cytotoxic activities and modes of binding to DNA, due to the different steric and/or electronic properties of the corresponding complexes.

Therefore, in this work, we studied the cytotoxic properties of six of these dinuclear iron compounds, and we found that all show IC_50_ values for tumor cell lines lower than that of cisplatin. Compound **10** presents the highest selectivity for tumor cells, and thus its mechanism of cell death induction was studied. We showed that compound **10** induces apoptotic cell death, likely acting over DNA, since in vitro it cuts both strands in the presence of reactive oxygen species (ROS). In addition, compound **10** is able to stop tumor cell migration on a highly metastatic cell line. Thus, it shows not only antitumor properties but also potential antimetastatic capacity.

## 2. Materials and Methods

### 2.1. Materials

Compounds **7–12** were synthesized using Schlenk techniques under nitrogen atmosphere and characterized following the methods described in the literature [15,16]. All reagents used were obtained from Sigma-Aldrich (Burlington, MA, USA), unless otherwise stated, and were used without further purification. Reagent-grade organic solvents were obtained from Carlo Erba (Barcelona, Spain).

The crystalline structures of the compounds studied in this work were previously resolved. Therefore, the crystallographic data, the details of the structure and solution, and the refinement procedures are described in the references [15,16]. An overview of these structures is given in Section 3.

### 2.2. Tested Compounds

Iron compounds were first reconstituted in DMSO and then diluted with sterile and bidistilled water to obtain 100 µM stock solutions with a concentration of 4% DMSO (Sigma-Aldrich, Burlington, MA, USA). Just before the experiments, these solutions were further diluted with sterile complete medium to obtain the desired final concentrations. In the cell proliferation experiments, cisplatin (Pfizer, Madrid, Spain) and carboplatin (Teva, Madrid, Spain) were included as controls.

### 2.3. Compounds Stability

From the stock solutions, each compound was adjusted to a final concentration of 10 μM with phosphate buffered saline (PBS), and, after incubating in the dark at 37 °C, their spectra, between 230 and 300 nm, were collected at different times (0, 24, 48 and 72 h) in a Perkin-Elmer Lambda BIO-20 UV–vis spectrophotometer (Perkin-Elmer, Waltham, MA, USA).

### 2.4. Cell Lines and Culture Conditions

NCI-H460 human lung carcinoma and OVCAR-8 human ovarian carcinoma cell lines were purchased from the National Cancer Institute-Frederick DCTD tumor cell line repository while CCD18-Co human colon fibroblasts were purchased from Celltec UB (Universitat de Barcelona, Barcelona, Spain). MDA-MB-231 human breast carcinoma cell line was purchased from the American Type Culture Collection (ATCC, Rockville, MD, USA).

NCI-H460, OVCAR-8, and MDA-MB-231 cell lines were routinely grown in 10% fetal bovine serum (FBS)-supplemented RPMI, 50 U/mL penicillin, 50 μg/mL streptomycin, and 2 mM L-glutamine. CCD-18Co cells were grown in 10% FBS-supplemented DMEM with 50 U/mL penicillin, 50 μg/mL streptomycin, and 2 mM L-glutamine. In all cases, cells were routinely grown at 37 °C in a humidified 5% CO_2_ atmosphere, and remained free of mycoplasma throughout the experiments. All reagents used for cell culture were from Lonza (Basel, Switzerland).

### 2.5. Cell Proliferation Assays

Cells were seeded into 96-well plates at densities of 1500 cells per well for OVCAR-8, 1900 for NCI-H460, and 4000 for CCD-18Co. After 24 h of incubation, they were treated for 72 h with different concentrations of the compounds (concentration range: 0.01 to 6 μM for OVCAR-8; 0.05 to 2 μM for NCI-H460; and 0.07 to 70 μM for CCD-18Co cells). When indicated, cells were also incubated with 30 µM of the ROS scavenger Manganese (III) 5,10,15,20-tetrakis (4-benzoic acid) porphyrin (MnTBAP) (Calbiochem, San Diego, CA, USA). Cell viability was measured using the MTT assay as previously described [17]. The IC_50_ value corresponds to the concentration of the assayed compound that is required to inhibit the cell proliferation by 50%, and was calculated by linear interpolation from the obtained growth curves. All data are reported as the mean ± standard error (SE) calculated from at least three independent experiments with three replicates in each.

### 2.6. DNA Interaction Analysis

The interaction of DNA with the compounds was monitored in an agarose gel electrophoresis as previously described [18]. Briefly, we mixed 0.5 μL of supercoiled 0.5 μg/μL pUC18 DNA (Thermo Scientific, Waltham, MA, USA) with either 25, 50, 75, or 100 µM of the compounds diluted in TE buffer (10 mM Tris-HCl, pH 7.6, 1 mM EDTA) in a final volume of 20 μL. Two other samples were included in the assays: a sample of pUC18 DNA alone as negative control and a sample of pUC18 DNA with 5 µg/mL cisplatin for comparison purposes. To assess the effect of ROS in mediating the interaction of the compounds with DNA, 1 µL of 30% H_2_O_2_ (*w*/*v*) was also added to the reaction. The samples were then incubated for 24 h at 37 °C and submitted to electrophoresis for 70 min at 100 V on a 0.8% agarose gel in 0.5x TBE buffer (89 mM Tris-borate, pH 8.3, 2 mM EDTA. Gels were stained with ethidium bromide (1 µg/mL in TBE) for 15 min and DNA bands were visualized under UV light.

### 2.7. Apoptosis Analysis

Quantitative analysis of apoptotic cell death caused by compound **10** treatment was carried out by flow cytometry using the Alexa Fluor 488 annexin V/PI Vybrant Apoptosis Assay Kit (Molecular Probes, Eugene, OR, USA) as previously described [19]. Briefly, NCI-H460 cells (7 × 10^4^ per well) and OVCAR-8 cells (8 × 10^4^ per well) were seeded into 6-well plates and then treated with compound **10** (0.64 μM for NCI-H460 and 1.65 μM for OVCAR-8) for 72 h in serum starved medium. After treatment, we collected attached and floating cells, washed them in cold PBS and stained them for 15 min in the dark with Annexin V-Alexa Fluor 488 and propidium iodide (PI) at room temperature. Cells were analyzed using NovoCyte Flow Cytometer (ACEA Biosciences, San Diego, CA, USA) and NovoExpress software (ACEA Biosciences, San Diego, CA, USA). At least 10,000 cells within the gated region were analyzed.

### 2.8. Cell Cycle Phase Analysis

NCI-H460 and OVCAR-8 cells were treated for 72 h with compound **10** (0.3 μM for NCI-H460 and 0.8 μM for OVCAR-8). Then, they were collected and fixed at −20 °C with 70% ethanol for at least 1 h. Fixed cells washed with cold PBS were resuspended in PBS at a cell density of 1–2 × 10^6^ cells/mL, treated at 37 °C for 30 min with ribonuclease A (100 μg/mL) and PI (40 μg/mL) (Molecular Probes, Eugene, OR, USA), and analyzed by flow cytometric. At least 10,000 cells within the gated region were analyzed on a NovoCyte Flow Cytometer (ACEA Biosciences, San Diego, CA, USA). Cell cycle distribution was analyzed using NovoExpress software (ACEA Biosciences, San Diego, CA, USA).

### 2.9. Flow Cytometric Analysis of ROS Generation

ROS generation after treatment with the compounds was analyzed using a carboxy-20, 70-dichlorodihydrofluorescein diacetate probe (carboxy-H_2_DCFDA) (Invitrogen, Waltham, MA). ROS species oxidize this molecule to green fluorescent dichlorofluorescein (DCF), allowing their detection inside the cells using a flow cytometer. A total of 24 h prior to treatments cells were seeded at 70,000 cells/well in 6-well plates in phenol red-free RPMI. Then, they were treated for 48 or 72 h at 37 °C with different concentrations of compound **10** (0.7, 1.5, or 3 μM for NCI-H460 cells and 2, 4.5, or 9 μM for OVCAR-8 cells) for 48 or 72 h at 37 °C. After treatments, cells were washed with PBS and incubated for 30 min at 37 °C in the dark with either 1 μM (NCI-H460 cells) or 0.5 μM (OVCAR-8 cells) carboxy-H_2_DCFDA diluted in PBS. Cells were then collected with phenol red-free trypsin and analyzed using a NovoCyte Flow cytometer. NovoExpress software (ACEA Biosciences, San Diego, CA, USA) was used to stablish the geometric mean fluorescence intensity of 10,000 cells.

### 2.10. Transwell Cell Migration Assay

MDA-MB-231 cells were seeded into the upper chamber of a 24-well transwell insert (8 µm pore size; Sarstedt, Nümbrecht, Germany) at a density of 12,000 cells/0.5 mL RPMI 1640 medium with 0.5% FBS and treated with 0.1 µM compound **10** (corresponding to an IC_20_). The lower chamber was filled with 0.5 mL RPMI 1640 medium with 10% FBS as a chemoattractant. The cells on the upper side of the inserts were removed with a cotton swab after 24, 48, or 72 h of incubation at 37 °C. Cells that migrated to the lower side of the inserts were fixed for 30 min with 4% paraformaldehyde and stained for 20 min at room temperature with 0.2% crystal violet solution. Ten randomly selected fields at 200× magnification were counted using an Olympus CKX41 inverted microscope (Tokyo, Japan).

### 2.11. Statistical Analysis

Results were analyzed using the Student’s *t* test. *p*-values < 0.05 were considered statistically significant. All statistical analyses were carried out with IBM SPSS Statistics 23 software for Windows (Armonk, NY, USA).

## 3. Results and Discussion

### 3.1. Structures of the Compounds and Stability

The synthesis and characterization of the compounds (**7–12**) have been previously reported [15,16]. The structures of molecular dinuclear iron compounds (**8–11**) containing the tetradentate ligands (**2–5**) (Figure 1) show the same arrangement for all the compounds, with an octahedral environment for the two iron atoms and the formation of the same isomer with *C_2h_* symmetry. Slightly distorted octahedrons were observed due to a steric effect when they present substituents in *ortho* position of the pyridine nitrogen atom. The disposition of the DBDOC backbones is the same for all the compounds and is not affected either by steric hindrance or by the electronic variations in the ligands [15].

In physiological conditions, the stability of all the compounds was assessed by monitoring for a period of 72 h the changes in their UV–vis spectra along the time (Appendix A displays the spectra for all the compounds assayed). No significant changes from the initial spectrum were obtained; thus, the compounds are stable in PBS at pH 7.4.

### 3.2. Cytotoxicity Assays

We evaluated the cytotoxicity of the iron compounds (**7–12**), with iminopyridine derivative ligands (Figure 1) and CH_3_CN as labile ligand, using two human carcinoma cell lines, NCI-H460 and OVCAR-8, and in non-tumor cells, CCD-18Co, after 72 h of exposure to them. All compounds exhibited cytotoxicity in a dose-dependent manner. Table 1 shows the IC_50_ values for all compounds, together with those of cisplatin and carboplatin, tested on the same cell lines.

As can be seen, all compounds are cytotoxic for all cell lines assayed, showing similar IC_50_ values for each cell line. Among the tumor cell lines, NCI-H460 is more prone to the action of iron compounds than OVCAR-8.

Table 1 shows that the cytotoxicity of all the tested iron compounds is from 2 to 6-fold higher than that of cisplatin for the assayed cell lines, and their IC_50_ values are about 30- to 90-fold lower than that of carboplatin for the assayed cell lines. For the non-tumor cells CCD-18Co, the cytotoxicity of the compounds compared to that of cisplatin and carboplatin is also higher, but with major variability depending on the compound. Table 1 also shows the selectivity index (SI) of all the compounds for both tumor cell lines compared to those of cisplatin and carboplatin. Interestingly, compounds **10** and **11** show the highest selectivity for cancer cell lines. That of compound **10** is even higher than that of cisplatin and carboplatin. The electronic effects of the substituents on position 6 may be important for the antiproliferative activity of the complexes and consequently for their selectivity values. Compounds **10** and **11** contain electron-donating groups at position 6 of the pyridine rings (-Me (**11**) and -OMe (**12**)). The donating character of these groups could induce a favorable interaction with the target molecules and even promoting the interaction through hydrogen bonding with these biomolecules.

### 3.3. Effect of Compounds on DNA

We tested whether the cytotoxicity for tumor cells of the entire set of compounds was due to their ability to interact with DNA. To this end, four concentrations of each compound were incubated at 37 °C for 24 h with cccDNA (plasmid pUC18). The integrity and topological conformation of the DNA were assessed using an agarose gel electrophoresis. Results are shown in Figure 1A. As can be observed, the position and intensity of the bands corresponding to the supercoiled form (CCC) and circular nicked form (OC) are identical to those of the negative control (plasmid pUC18 alone); therefore, the compounds are not able to interact with DNA at all the concentrations assayed. On the other hand, the positive control, cisplatin, promotes the described effect on DNA, i.e., the migration of the CCC form decreases until it co-migrates with the OC form to reach the coalescence point [20]. We performed the same assay in the presence of H_2_O_2_, an initial activator of other ROS compounds [18]. As can be seen (Figure 1B), in the presence of H_2_O_2_ all the compounds promote the appearance of a band, which is formed when both DNA strands of plasmid pUC18 are cut, named the linear form (L). Therefore, in the presence of ROS all the compounds cut the DNA. Moreover, the highest concentrations of compounds **8**, **10,** and **12** promote total DNA degradation, visualized by the disappearance of all the bands. Taking into account that dinuclear or multi-nuclear compounds, including those of platinum, act on DNA in a different way than cisplatin [8], the results are not surprising. Moreover, it has been described that some dinuclear iron compounds present high nuclease activity [12]. The fact that all the compounds produce the linearization of pUC18 substrate, even at the lowest concentrations, suggests that this activity is enhanced in our compounds. Nevertheless, according to the results of their ability to cleave DNA, we cannot explain the higher selectivity of compound **10** for tumor cells compared to the other compounds. The high selectivity of this compound makes it an interesting potential antitumor agent; thus, we have further characterized its cytotoxic effect.

### 3.4. Compound **10** Induces Apoptosis in the Tumor Cell Lines

One of the desired characteristics of anticancer agents is their ability to kill cancer cells through the induction of apoptosis. We investigated by flow cytometry whether the most selective compound, **10**, triggers apoptosis on two tumor cell lines, NCI-H460 and OVCAR-8, using a concentration equal to the IC_50_ value for each cell line (0.64 and 1.65 µM, respectively) after 72 h of exposure. The flow cytometry results of the treated cells, once stained with Annexin V-Alexa Fluor 488 and propidium iodide (PI), are shown in Table 2. As can be seen, when both cell lines are treated with compound **10,** an increase of early and late apoptotic cells is observed related to the non-treated cells. On the other hand, in the OVCAR-8 cell line the percentage of cells treated with compound **10** that are in early and late apoptosis is similar, while in NCI-H460 the percentage of late apoptotic cells is significantly higher than that of early apoptotic cells. Globally, the percentage of cells that enter in apoptosis is about 82% and 66% for NCI-H460 and OVCAR-8 cells, respectively. This is in agreement with the increased sensitivity of the NCI-H460 cell line to compound **10** compared to that of the OVCAR-8 cell line. Interestingly, the percentage of necrosis induced by compound **10** is negligible in both cell lines, suggesting that it would not induce inflammation. This result is beneficial for its potential use as an antitumor drug.

### 3.5. Effects of Compound **10** on the Cell Cycle Phase Distribution

The two main processes that account for the inhibition of cell growth are apoptosis and cell cycle arrest. Therefore, we also assessed, by flow cytometry, the effect of compound **10** on NCI-H460 and OVCAR-8 cell cycle phase distribution. Figure 2 presents the effects of compound **10** on cell cycle phase distribution for the tumor cell lines, NCI-H460 and OVCAR-8, compared to untreated cells. After 72 h of exposure to compound **10** we observed that NCI-H460 cells do not show significant changes in the cell cycle phase distribution compared to untreated cells, indicative that in this cell line compound **10** does not induce cell cycle arrest, i.e., it behaves as a cell-cycle independent agent (Figure 2A). Compound **10** causes a shift in the cell population from the G_0_/G_1_ phase to the S and G_2_/M phases in OVCAR-8 cell line when compared to control, which is indicative of cell cycle arrest in these final phases of cell cycle (Figure 2B). Compound **10** exerts a cytotoxic effect in the OVCAR-8 cell line, and likely presents an antiproliferative effect.

### 3.6. Compound **10** Triggers ROS Generation

Agarose gel assays prompted us to evaluate whether ROS play a significant role in the cytotoxicity induced by compound **10**. Using flow cytometry, we assessed whether the use of this compound augmented the levels of ROS in NCI-H460 and OVCAR-8 cell lines. Compound **10**, at different concentrations, was added to cells, and the treatment lasted for 48 or 72 h. Then, cells were labelled with carboxy-H_2_DCFDA. Figure 3 shows that, in both cell lines, ROS levels rise with time and concentration. At 48 h, the ROS production by compound **10** is significantly higher in the OVCAR-8 cell line than in the NCI-H460 cell line. The OVCAR-8 cell line is more resistant to compound **10** than the NCI-H460 cell line, in spite of the higher ROS values produced by the treatment. To further prove the involvement of ROS in the cytotoxicity of compound **10**, we tested whether the addition of a reducing agent could affect its cytotoxicity on the OVCAR-8 cell line. To this end, we measured cellular viability, using the MTT assay, of the OVCAR-8 cell line treated with different concentrations of compound **10** in the presence and absence of the reducing agent MnTBAP. This reagent offsets ROS production in cells mainly by its ability to scavenge superoxide species [21]. Figure 4 shows that MnTBAP reduces the antiproliferative effect of compound **10** at the concentrations assayed. Thus, the cytotoxicity induced by compound **10** clearly involves ROS production. However, the lower sensitivity of the OVCAR-8 cell line to compound **10,** and its higher levels of ROS production once treated with this compound, suggests that ROS production is likely not the only antiproliferative mechanism triggered by compound **10**.

### 3.7. Compound **10** Precludes Migration of Tumor Cells

Together with the induction of apoptosis, another interesting feature of antitumor agents is their ability to stop cell migration to avoid metastasis. We investigated whether compound **10** could arrest cell migration using a transwell assay. NCI-H460 and OVCAR-8 cell lines were not adequate to carry out this experiment because they have a very low capacity for migration; therefore, we used the highly invasive breast cancer cell line MDA-MB-231. Firstly, we investigated the cytotoxicity of compound **10** on this cell line and we found that its IC_50_ was 0.25 µM, even lower than that observed in the NCI-H460 and OVCAR-8 cell lines. Interestingly, the IC_50_ value for this cell line is about 250-fold lower than that described for cisplatin [22]. Figure 5 shows that compound **10** significantly reduces the migration of the MDA-MB-231 cancer cell line.

## 4. Conclusions

We characterized the antitumor properties of a family of new dinuclear Fe(II) complexes containing iminopyridine ligands based on a methanodibenzodioxocine (DBDOC) backbone. All compounds show IC_50_ values less than those of cisplatin and carboplatin for the cancer cell lines assayed. In addition, in the presence of H_2_O_2_, all compounds interact with DNA, promoting double strand breaks. We have found that, among them, compound **10,** which contains an electron-donating group at position 6 of the pyridine rings (-Me), shows the highest selectivity for tumor cells, which is even higher than that of cisplatin and carboplatin. Compound **10** induces cell death of tumor cells by apoptosis, but has an effect on cell cycle phases in a cell-dependent manner. This apoptosis seems to be promoted by DNA strand breaking, a process that is helped by ROS generation. Interestingly, compound **10** also inhibits tumor cell migration. Thus, it behaves not only as an antitumor agent, but also as a potential antimetastatic drug. In conclusion, we found a dinuclear iron compound that is a likely alternative to the present metal-based antitumor agents.

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
