# Peer review of "Dinuclear Iron Complexes of Iminopyridine-Based Ligands as Selective Cytotoxins for Tumor Cells and Inhibitors of Cancer Cell Migration"

_pharmaceutics, 2022, doi:10.3390/pharmaceutics14122801_

Round 1

Reviewer 1 Report

1- The introduction section must be reformulated and rewrite to become more informative concerning the topic of article.

2- The references which were cited in the article are old and authors must be updated them.

3- The experimental section must be give an comprehensive idea about the interpretation of structures of iron complexes which synthesized.

4- Some figures must be need to become more clear resolution and also put in article with color feature like figures 1, 3, 4 and 5.

5- The results and discussion section need more assignments, authors were written in summarized statement.

6- Authors must be check in all article to defined the abbreviations.

7- Section 3.1 and 3.2 must be rewrite to become more clear discussions.

Author Response

Thank you very much for your comments. We have addressed them as follows:

1, 2, 5 and 7. All the text has been revised and reference 1 has been changed to a more recent one. In our opinion, the rest of the references are not old but selected to place the reader on the main results achieved in the field of iron compounds and also dinuclear compounds as antitumor drugs.

  1. The experimental section has also been revised. All the procedures and protocols described are set-up in our laboratory and are written to be reproducible.
  2. The figures that in the previous version were not in color are now redrawn and all of them are in color and with a good resolution.
  3. According to the reviewer’s comment, all abbreviations are defined the first time that they appear in the text.

All the changes are marked in the revised version.

Reviewer 2 Report

Review on manuscript pharmaceutics-1953187

“Dinuclear Iron Complexes of Iminopyridine Based Ligands as 2 Selective Cytotoxins for Tumor Cells and Inhibitors of Cancer 3 Cell Migration”

by J. Castro, M. Bravo, M. Albertí, A. Marsal, M.J. Alonso-De Gennaro, O. Martínez-Ferraté, C. Claver, P.W.N.M. van Leeuwen, I. Romero, A. Benito, M. Vilanova

Recommendation

Accept, a minor revision is needed.

Overall statement on the manuscript

This research article demonstrates an important study about the formation of several iron(II) compounds with iminopyridine-based ligands with several (or ten-) fold higher anti-cancer efficiency in comparison with cisplatin or carboplatin drugs. Among the introduced dinuclear iron complexes of iminopyridine based ligands, the synthesized [Fe2(4)2(CH3CN)4]( BF4)4, 10 compound exhibits the highest anti-cancer selectivity through the mechanism of apoptosis induction, interfering with the cell cycle phase distribution and enhancing cytotoxic effect being linked to ROS generation, thereby promoting the anti-metastatic effects. This work is entirely comprehensive, reproducible and has a logical sequence. The experimental results are fair and their interpretation is trustworthy. The statements in the article are correctly derived and proved by cited references. Therefore I recommend this article for publication.

I have one comment about the order of the cited references and probable missed references and one question about the UV-Vis spectra. In the manuscript of this article the reference 20 comes after the reference 16 (p.7) and the reference 18 (p.8) comes after reference 20. I couldn’t find the cited references 17 and 19. Why does the UV-Vis absorption intensity of synthesized complexes decreases after 72 h of incubation?

Best regards

Author Response

Thank you very much for your comments. We have addressed them as follows:

  1. Regarding references, please note that reference 20 (line279) comes after reference 19 (line 180), which is cited in section 2.7. References 17 and 18 are cited in sections 2.5 (line 160) and 2.6 (line 166), respectively. Thus, all the references are cited in the correct order.
  2. In relation to the intensities of the UV-absorption spectra of complexes, please note that the differences are minimal and the small increases or decreases in intensity along time are negligible and do not reflect any changes in the compounds related to the their stability, which is the object of this study.

Reviewer 3 Report

The article has the main contribution to other than chemits.

Author Response

We agree with the reviewer that the paper is focused on the biological characterization (antitumor and antimetastatic activities) of the dinuclear iron compounds while their synthesis and structural characterization was done earlier (ref 15 and 16) as indicated in the penultimate paragraph of the Introduction section.

Reviewer 4 Report

The manuscript ID pharmaceutics-1953187 reveals new antitumor mechanisms exhibited by a class of dinuclear iron complexes. The authors proves that certain ligands confer to the diiron complexes excellent inhibitory capacity and selectivity towards the tumor cells. The study design and the methodology are appropriate, the conclusion are sustained by several in vitro experiments on tumor cell lines. The authors rigorously described all experimental results (cytotoxicity, apoptosis induction, cell cycle arrest, migration hindrance but also lack of interaction with plasmid DNA, and so on) seeking the reasons why compound 10 anticancer properties are so encouraging. In all chapters, the reader could recognize proofs of reproducibility (see below comment for page 7, as an example).  

I recommend publishing the paper, with some minor changes.

Page 2, last paragraph: instead "some of us previously synthesized..." authors may write: a series of dinuclear iron compounds were synthesized (..) Please describe which ligand correspond to complexes 7-12, since this paragraph precede page 6, chapter 3.1 and even on page 6 a disambiguation is needed. The name of complex 8 contains empty brackets. Schemes 1 and especially 2 requires a larger figure caption, with relevant details, such color of Fe, B, F; the software used to generate the image should be inserted, and please mention in the figure caption references 15 and 16. If a cif file is deposited to CCDC or elswhere, please insert the link.

Page 3, lines 101-108: some of the sentences match better to the discussion/ results

In the Material and methods section: for all providers, city/country should be inserted.

Chapter 6.2 "appropriate aliquots of the compounds"- all the used concentrations should be mentioned here; same for the rest of the subchapters

Page 6, 3.1: DBDOC appears here first, please insert its explanation

Page 7, row 256: for readers who are not familiar with IC50 meaning, a very short explanation about lower IC50/higher cytotoxicity correlation might be useful.

As a remark: The authors  compare even the IC50 values of standard platinum drugs with the earlier results- in this case the differences could be attributed to different details in protocols/ cell growth conditions/reagents therefore they are accurate and can serve as comparison with the diiron compounds IC50.   

Page 4, in conclusions chapter: it could be useful to describe whether the iminopyridine ligands or CH3CN labile ligands influences the cell growth inhibition induced by compounds 7-12.

Author Response

Thank you very much for your comments. We have addressed them as follows:

  1. The sentence "some of us previously synthesized..." has been changed as indicated “a series of dinuclear iron compounds were synthesized”
  2. The ligands are indicated in Scheme 1. To disambiguate, the legend of Scheme 1 has been extended to indicate which ligand corresponds to each compound, as follows:

Scheme 1. Drawing of the imino- pyridine ligands used in the synthesis of the complexes of general formula [Fe2L2(CH3CN)4](BF4)4 , studied in this work. (L=1 for compound 7; L=2 for compound 8; L=3 for compound 9; L=4 for compound 10; L=5 for compound 11 and L=6 for compound 12).

In addition, in section 3.1 it has been added, “(Scheme 1)”, after “tetradentate ligands (2-5)

  1. The name of complex 8 has been corrected. We apologize for this typing error.
  2. Figure legends of schemes 1 and 2 have been modified according to the reviewers’ indications: color of Fe, B, F; the software used to generate the image, and we cite ref. 15 and 16 and the CCDC file.

  1. Page 3, lines 98-105. We agree with the reviewer that some sentences are the results of this work. Nevertheless, they are just to focus the reader on the achievements of the work. We consider that the sentences are informative for readers.

  1. In the Material and methods section all providers, city/country have been indicated according with the reviewer’s comment

  1. Paragraph 2.6. The final concentrations of the compounds are now indicated: “(Briefly, we mixed 0.5 μl of supercoiled 0.5 μg/μl pUC18 DNA (Thermo Scientific, Waltham, MA, USA) with either 25, 50, 75 or 100 µM of the compounds diluted in TE buffer (10 mM Tris-HCl, pH 7.6, 1 mM EDTA) in a final volume of 20 μl.)”. In the rest of the paragraphs, they are also indicated.

  1. The abbreviation DBDOC appears before page 6 (line 243). It appears in the introduction section, page 2 (line 87) and there the abbreviation is indicated “methanodibenzodioxocine”

  1. The meaning of the IC50 value is indicated in the footnote of Table 1 (line 248) which is very close to the first time that it appears in the Results and Discussion (line 245). In addition, the meaning of this value is also explained in section 2.5 (line 160) of Materials and Methods. We consider that it is not necessary to repeat the explanation.

  1. We agree with the remark of the reviewer about the comparison with the cytotoxicity of the iron compounds with cis-platin and carboplatin.

  1. We also agree with his/her suggestion that it could be useful to know if the iminopyridine ligands have any effect on the cell growth inhibition. Unfortunately, we do not have all the free ligands to test their effect. Nevertheless, we have tested the cytotoxicity of ligands 1-3 and 6 on NCI-H460 and OVCAR-8 cell lines and the IC50 values are clearly higher than that of their corresponding compounds. Therefore, we believe that the coordination of the ligands to iron increases their antitumor activity. However, since it has not been possible to carry out this assay with all the ligands we have not included it in the results section. Regarding the labile ligand, acetonitrile, at the final concentration added to the cells after the preparation of the compound, its effect on the cell viability is minor if any. In other works (please see ACS Omega 2020, 5, 13984−13993; https://dx.doi.org/10.1021/acsomega.0c01384) it is described that a concentration of acetonitrile of 40 µM has no effect on cell viability. We are aware that the cell lines assayed are different that the ones used in our work, however, taking into account the low IC50 values that we get for our compounds (Table 1) we are sure that the acetonitrile contribution to the cytotoxicity is negligible. Moreover, the concentrations equivalent to the IC50 values of tumour cell lines do not have effect on the non-tumour cell line assayed that present IC50 values remarkable higher.

Round 2

Reviewer 1 Report

Accept